# Tannylated Calcium Carbonate Materials with Antacid, Anti-Inflammatory, and Antioxidant Effects

**DOI:** 10.3390/ijms22094614

**Published:** 2021-04-28

**Authors:** Sung-Yun Jung, Heamin Hwang, Han-Saem Jo, Somang Choi, Hak-Jun Kim, Sung-Eun Kim, Kyeongsoon Park

**Affiliations:** 1Department of Systems Biotechnology, Chung-Ang University, Gyeonggi 17546, Korea; jsy0035@gmail.com (S.-Y.J.); heamin1997@naver.com (H.H.); luchiatkfkd@naver.com (H.-S.J.); 2Department of Orthopedic Surgery and Nano-Based Disease Control Institute, Korea University Guro Hospital, #148, Gurodong-ro, Guro-gu, Seoul 08308, Korea; chlthakd1029@naver.com (S.C.); dakjul@korea.ac.kr (H.-J.K.)

**Keywords:** calcium carbonate (CaCO_3_), tannic acid (TA), antacid, anti-inflammation, ROS scavenging activity, chondrocytes

## Abstract

Calcium carbonate (CaCO_3_)-based materials have received notable attention for biomedical applications owing to their safety and beneficial characteristics, such as pH sensitivity, carbon dioxide (CO_2_) gas generation, and antacid properties. Herein, to additionally incorporate antioxidant and anti-inflammatory functions, we prepared tannylated CaCO_3_ (TA-CaCO_3_) materials using a simple reaction between tannic acid (TA), calcium (Ca^2+^), and carbonate (CO_3_^2−^) ions. TA-CaCO_3_ synthesized at a molar ratio of 1:75 (TA:calcium chloride (CaCl_2_)/sodium carbonate (Na_2_CO_3_)) showed 3–6 μm particles, comprising small nanoparticles in a size range of 17–41 nm. The TA-CaCO_3_ materials could efficiently neutralize the acid solution and scavenge free radicals. In addition, these materials could significantly reduce the mRNA levels of pro-inflammatory factors and intracellular reactive oxygen species, and protect chondrocytes from toxic hydrogen peroxide conditions. Thus, in addition to their antacid property, the prepared TA-CaCO_3_ materials exert excellent antioxidant and anti-inflammatory effects through the introduction of TA molecules. Therefore, TA-CaCO_3_ materials can potentially be used to treat inflammatory cells or diseases.

## 1. Introduction

Inorganic minerals have been widely used in drug delivery systems for biomedical applications [1]. Calcium carbonate (CaCO_3_), an inorganic biomineral, has been used as an antacid agent. It can be orally administered as a tablet, chewable tablet, capsule, or liquid. Furthermore, it has been used for the controlled and sustained delivery of chemical drugs [2,3,4], photosensitizers [5], and proteins [6,7] because of its biocompatibility and slow biodegradation [8]. CaCO_3_ is stable at physiological pH, but can be dissociated under acidic conditions [9,10]. Owing to their pH sensitivity, CaCO_3_-based delivery systems concentrate drugs into targeted cancer tissues within the acidic tumor microenvironment (TME) [4,5,10]. In addition, these systems react with protons (H^+^) to neutralize the acid [11,12]. CaCO_3_ can generate carbon dioxide (CO_2_) in the acidic TME, and this gas-generating property has extended its application as an ultrasound contrast agent for cancer imaging [5,10]. These previous studies have revealed the pH sensitivity, CO_2_ gas generation, and antacid properties of CaCO_3_ materials.

Polyphenol tannic acid (TA) is composed of five digalloyl ester groups that have been linked to a central glucose molecule, and exhibits various biological properties, such as antibacterial, anti-inflammatory, antioxidant, and anticancer activities [13,14]. Previous studies have revealed the TA-mediated scavenging of free radicals, leading to the inhibition of lipid oxidation and radical-mediated DNA damage [13,15,16]. Given their ability to undergo multiple interactions with various biomacromolecules (i.e., nucleic acids, peptides, proteins, and polysaccharides) through electrostatic and hydrogen bonding and/or hydrophobic interactions [17,18,19,20,21], TA–macromolecular complexes can be easily produced and used as surface modifiers on organic and inorganic substrates [22,23]. Thus, the TA-modified polymeric hydrogels and scaffolds greatly enhance anti-inflammatory effects in vitro or protect cells under reactive oxygen species (ROS) environments [23,24]. Moreover, as TA can coordinate with metal ions, it can be used to synthesize inorganic Ag and Au nanomaterials [25]. Our group recently prepared oxygen-generating calcium peroxide using TA through coordination between the catechol moieties of TA and calcium ions [26].

Based on these previous reports, we propose that the introduction of TA into CaCO_3_ materials can endow them with anti-inflammatory and antioxidant functions. As CaCO_3_ materials exert no anti-inflammatory or antioxidant functions, herein we fabricated tannylated CaCO_3_ (termed as TA-CaCO_3_) materials via the coordination of TA to calcium (Ca^2+^) ions and further nucleation of CaCO_3_ using carbonate ions (CO_3_^2−^). We performed their physicochemical characterization and evaluated their antacid and antioxidant effects using colorimetric methods. In addition, we validated their in vitro antioxidant and anti-inflammatory properties in chondrocytes under inflammatory and ROS conditions.

## 2. Results and Discussion

### 2.1. Preparation of TA-CaCO_3_ Materials

TA-CaCO_3_ materials were synthesized by combining equimolar amounts of Ca^2+^ and CO_3_^2−^ at different concentrations with a fixed molar concentration of TA under constant stirring, as depicted in Figure 1. Given the tendency of TA to coordinate with metal ions [25,27,28], TA interacts with Ca^2+^ ions and facilitates the nucleation of CaCO_3_ to form TA-CaCO_3_ nanoparticles, which then agglomerate into microparticles due to the interactions between TA-CaCO_3_ nanoparticles.

### 2.2. Characterization of TA-CaCO_3_ Materials

The morphologies of the synthesized TA-CaCO_3_ materials were examined using scanning electron microscopy (SEM). As shown in Appendix A, at 25 molar ratios of calcium chloride (CaCl_2_)/sodium carbonate (Na_2_CO_3_), aggregates comprising small TA-CaCO_3_ particles were predominantly observed, and very small amounts of micron-sized spherical TA-CaCO_3_ particles were formed. More spherical TA-CaCO_3_ microparticles were gradually observed as the molar ratio of CaCl_2_/Na_2_CO_3_ was increased to 75. Interestingly, although spherical TA-CaCO_3_ microparticles were still observed when the molar ratios of TA and CaCl_2_/Na_2_CO_3_ were 100 or 150, more irregular and broken TA-CaCO_3_ particles were detected. These results suggested that small TA-CaCO_3_ particles were formed at lower molar ratios of CaCl_2_/Na_2_CO_3_ and that spherical and broken TA-CaCO_3_ particles were more common at higher molar ratios of TA and CaCl_2_/Na_2_CO_3_. Based on SEM images, 1:75 TA-CaCO_3_ particles were selected for further experiments because they produced more spherical TA-CaCO_3_ microparticles than other TA-CaCO_3_ particles.

Next, we investigated the particle size of the 1:75 TA-CaCO_3_ materials and found it to range from approximately 3 to 6 μm (Figure 2a and Appendix A). Interestingly, the magnified SEM images revealed the presence of small particles on the surface of the 1:75 TA-CaCO_3_ materials. These individual small particles ranged from 17 to 41 nm in size, and were approximately 26.18 ± 4.6 nm in diameter and spherical in shape (Figure 2b). SEM images revealed that the micron-sized TA-CaCO_3_ materials consisted of small TA-CaCO_3_ nanoparticles, probably owing to agglomeration following interactions between individual small nanoparticles.

The preparation of TA-CaCO_3_ materials was confirmed using an SEM coupled with energy-dispersive (SEM-EDS) X-ray spectroscopy, inductively coupled plasma optical emission spectrometry (ICP-OES), Fourier transform infrared (FT-IR) spectroscopy, X-ray diffraction (XRD) patterns, and X-ray photoelectron spectroscopy (XPS). The EDS mapping and spectrum data revealed the presence of carbon, oxygen, and calcium on the surface of TA-CaCO_3_ materials (Figure 2c,d). Consistent with a previous report [29], the EDS spectrum showed the peaks of Kα (0.277 eV) corresponding to carbon, Kα (0.523 eV) corresponding to oxygen, and Kα (3.691 eV) and Lα (0.341 eV) corresponding to calcium. ICP-OES analysis showed that 0.1 mg of 1:75 TA-CaCO_3_ contained 18.6 μg of TA and 81.4 μg of CaCO_3_. The FT-IR spectrum of commercial CaCO_3_ showed the characteristic vibrations of carbonate ions (at 1805, 1410, 1090, and 874 cm^−1^) (Figure 3), as previously reported [30,31]. The preparation of TA-CaCO_3_ materials was confirmed by the presence of the main asymmetric vibrations at 1460 and 1410 cm^−1^. However, the symmetric vibration at 725 cm^−1^ disappeared, implying that TA-CaCO_3_ has an amorphous structure. Meanwhile, TA-CaCO_3_ showed characteristic peaks at 3300–3600 cm^−1^ (O-H stretching), 1445 cm^−1^ (C-C stretching of benzene ring and methylene; C-O stretching of phenolic), and 755 cm^−1^ (C-H torsion of benzene ring) [32], indicative of the presence of TA on the material.

Next, commercial CaCO_3_ and synthesized TA-CaCO_3_ crystal phases were identified via XRD analysis (Figure 4a). CaCO_3_ exhibited the characteristic peaks, such as the plane of the calcite at 29.3° (104), and the calcite crystal faces at 23.02° (012), 35.9° (110), 39.4° (113), and 43.1° (202), respectively. Meanwhile, the diffraction peaks of TA-CaCO_3_ were observed at 24.8° (110), 27.08° (112), 32.7° (114), 43.8° (300), 49.1° (304), and 50.08° (118), respectively, and these diffraction peaks were consistent with the vaterite crystal faces [33,34]. Based on XRD data, we think that the prepared TA-CaCO_3_ materials are the vaterite form of calcium carbonate.

XPS data revealed that CaCO_3_ and TA-CaCO_3_ showed Ca, O, and C signals (Figure 4b). The binding energy peaks of these two materials appeared O1s at 531 eV, Ca2s at 441 eV, two Ca2p at 351 and 347.2 eV, and two C1s peaks at 289.3 eV (CO_3_ in the CaCO_3_ surface) and 284.6 eV (adventitious carbon peak), respectively. These data demonstrated the successful synthesis of TA-CaCO_3_, and the synthesized TA-CaCO_3_ materials are vaterite calcium carbonate.

### 2.3. Antacid Effects of TA-CaCO_3_

CaCO_3_ exists as a stable crystalline solid at physiological pH, but can be dissociated into ionic species at or below weakly acidic pH [5,9]. Under acidic pH, CaCO_3_ neutralizes acids by reacting with the proton (H^+^) [11,12], and it has been used as an acid neutralizer [35].

To verify the antacid effects of TA-CaCO_3_ materials, commercial CaCO_3_ and TA-CaCO_3_ were dispersed in phosphate-buffered saline (PBS; physiological pH = 7.4) and simulated gastric fluid (SGF, pH 1.5) containing bromothymol blue (BTB). The color and absorption changes of BTB were monitored before and after the reaction, because BTB is a useful acid/base indicator to distinguish the acidity, neutrality, and alkalinity of an aqueous solution [36,37]. As shown in Figure 5a and Appendix A, the aqueous BTB solution without commercial CaCO_3_ and TA-CaCO_3_ exhibited a blue color at pH = 7.4 and turned to a yellowish color in the presence of SGF (pH = 1.5). CaCO_3_ and TA-CaCO_3_ showed a deep blue color of BTB at pH = 7.4, indicating the slight pH increases of the solution following the degradation of CaCO_3_ and TA-CaCO_3_, even at pH = 7.4. In contrast, the yellowish BTB solution at pH = 1.5 turned to a bluish green color after the reaction with CaCO_3_ and TA-CaCO_3_, indicating that the pH of the solution increased to a nearly neutral pH (approximately 7). Consistent with these color changes, the λ_max_ shift of BTB occurred from 615 nm at pH = 7.4 to 433 nm under SGF (pH = 1.5) (Figure 5b). However, the absorptions of both CaCO_3_ and TA-CaCO_3_ groups increased at 615 nm, while the λ_max_ of BTB at pH = 1.5 was blue-shifted from 433 nm to 403 nm, implying the pH increases of the solutions after reacting with two kinds of CaCO_3_ materials. Meanwhile, TA-CaCO_3_ showed significantly higher absorbance below 400 nm at both pH = 1.5 and pH = 7.4, indicating the presence of TA in the solutions. Furthermore, TA-CaCO_3_ had a lower absorbance than commercial CaCO_3_ at 615 nm. This is attributed to the fact that TA-CaCO_3_ contained a lower amount of CaCO_3_ than commercial CaCO_3_.

### 2.4. Antioxidant Effects of TA-CaCO_3_

The antioxidant effects of TA-CaCO_3_ were determined using the stable free radical 2,2-diphenyl-1-picrylhydrazyl (DPPH) [38]. As shown in Figure 6, the DPPH solution showed a deep violet color, with an absorbance at approximately 520 nm, owing to the delocalization of the spare electron over the molecule [38]. CaCO_3_-treated DPPH solution also had a deep violet color and an absorption band at around 520 nm, indicating that CaCO_3_ had no antioxidant effect. In contrast, TA-CaCO_3_-treated DPPH solution turned yellowish in color and lost its absorbance at 520 nm. This loss of violet color might be attributed to the presence of TA molecules within TA-CaCO_3_ because TA molecules have an effective radical-scavenging activity, as its hydroxyl groups easily reduce the free radicals of DPPH [39,40]. These data imply that TA-CaCO_3_ materials have antioxidant properties and effective ROS-scavenging activity.

### 2.5. In Vitro Anti-Inflammatory Effects of TA-CaCO_3_ in Inflamed Chondrocytes

The cytotoxicity study of TA-CaCO_3_ was examined against normal chondrocytes. Appendix A revealed that chondrocytes maintained their cell viabilities above 90% up to a 100 μg/mL concentration, indicating that TA-CaCO_3_ is non-toxic.

In addition to antioxidant effects, TA molecules also exhibit anti-inflammatory properties [41]. We examined the in vitro anti-inflammatory effects of TA-CaCO_3_ by analyzing the mRNA expression of pro-inflammatory factors, such as cyclooxygenase-2 (COX-2), interleukin (IL)-1β, IL-6, matrix metalloproteinase (MMP)-3, MMP-13, and tumor necrosis factor (TNF)-α in lipopolysaccharide (LPS)-stimulated chondrocytes, because LPS treatment leads to the overexpression of these pro-inflammatory factors [41,42,43].

As shown in Figure 7, LPS significantly upregulated the mRNA expression of all pro-inflammatory cytokines on day three compared to the control treatment. However, the mRNA levels of these pro-inflammatory cytokines were efficiently suppressed following treatment with TA-CaCO_3_ in a dose-dependent manner. These results indicate that TA-CaCO_3_ materials are highly effective in suppressing pro-inflammatory factors in LPS-stimulated chondrocytes. Consistent with previous studies, the present study also supports the anti-inflammatory effects of TA-based materials [24,44].

### 2.6. In Vitro Antioxidant and Protective Effects of TA-CaCO_3_ in H_2_O_2_-Treated Chondrocytes

The exogenous treatment of cells with H_2_O_2_ results in the induction of oxidative stress and produces intracellular ROS [45,46]. To demonstrate the antioxidant activities of TA-CaCO_3_ at the cellular level, H_2_O_2_ (300 μM)-pretreated chondrocytes were incubated with different concentrations of TA-CaCO_3_ extracts, and the intracellular ROS level was detected from the fluorescence signal of 2′,7-dichlorodihydrofluorescein diacetate (DCFDA) using a confocal laser scanning microscope. As shown in Figure 8a, no fluorescence was observed in normal cells, whereas a strong signal was detected in cells treated with 300 μM of H_2_O_2_ alone. The treatment of the extracts containing different concentrations of TA-CaCO_3_ led to a remarkable decrease in fluorescence intensities, indicative of the excellent scavenging of intracellular ROS by TA-CaCO_3_.

Previous studies have reported that H_2_O_2_ can be toxic to cells because it produces hydroxyl radicals [45,46]. Substances with antioxidant properties prevent cell damage and protect the human body from free radicals or ROS by supplying electrons from antioxidants to the damaged cells [47,48]. Based on these facts, we investigated whether TA-CaCO_3_ with antioxidant properties is effective in mediating cellular protection by measuring the viability of chondrocytes treated with 300 μM of H_2_O_2_ (Figure 8b). In comparison with the control group, the chondrocytes cultured in 300 μM of H_2_O_2_ showed a significant decrease in viability owing to the oxidative damage to cellular components [49]. However, the viability of the cells treated with different concentrations of TA-CaCO_3_ significantly increased in a concentration-dependent manner, and was much higher than that of the chondrocytes treated with 300 μM of H_2_O_2_ alone. Consistent with our previous studies showing that substances with antioxidant properties could protect the cells against ROS environments [23,50], TA-CaCO_3_ effectively protected the cells and prevented cellular damage under ROS conditions. This protective effect might be associated with the effective radical-scavenging activity of TA molecules within TA-CaCO_3_ materials.

## 3. Materials and Methods

### 3.1. Materials

Calcium chloride dihydrate (CaCl_2_·2H_2_O, abbreviated as CaCl_2_), sodium carbonate monohydrate (Na_2_CO_3_·H_2_O, abbreviated as Na_2_CO_3_), TA, potassium bromide (KBr), BTB, DPPH, sodium hydroxide (NaOH), 26% sodium chloride (NaCl) solution, and hydrochloric acid (HCl) were purchased from Sigma-Aldrich (St. Louis, MO, USA). Methanol (MeOH, 99.5%) and absolute ethanol (EtOH, 99.9%) were provided by Samchun (Pyeongtaek, Korea) and DUKSAN (Ansan, Korea), respectively. Commercial CaCO_3_ nanopowder (purity: 98%) was supplied by US Research Nanomaterials (Houston, TX, USA), and PBS by Lonza (Walkersville, MD, USA). SGF (pH = 1.5) was prepared using ultra-pure water (99.46% *v*/*v*), concentrated HCl (0.23% *v*/*v*), and 26% NaCl solution (0.21% *v*/*v*). To prepare the BTB solution, 10 mg of BTB was dissolved in 1 mL of 4% NaOH solution, and treated with 2 mL of 70% EtOH (2 mL) and 100 mL of ultra-pure water. Finally, a 10% NaOH solution was added dropwise to the BTB solution when the color of the BTB solution changed to blue.

### 3.2. Preparation of TA-CaCO_3_

For the preparation of TA-CaCO_3_ materials, TA (0.0294 mmol) was added to ultra-pure water (5 mL, pH = 7.0) and stirred to dissolve at 800 rpm for 1 h. Different moles of CaCl_2_ (0.735, 1.47, 2.20, 2.94, and 4.41 mmol) were dissolved in ultra-pure water (3 mL, pH = 7.0). The prepared CaCl_2_ solutions were added to the TA solution and stirred at 800 rpm for 2 h. In addition, different moles of Na_2_CO_3_ (0.735, 1.47, 2.20, 2.94, and 4.41 mmol) were dissolved in ultra-pure water (pH = 7.0) and added dropwise to the mixture solution of TA and CaCl_2_. The mixture was stirred at 800 rpm for 24 h, and the resulting solution was centrifuged at 1200 rpm for 5 min and washed with ultrapure water. The collected TA-CaCO_3_ materials were freeze-dried for two days. TA-CaCO_3_ materials synthesized at different molar ratios of TA (molar ratio: 1), CaCl_2_ (molar ratios: 25, 50, 75, 100, and 150), and Na_2_CO_3_ (molar ratios of 25, 50, 75, 100, and 150) were named as follows: 1:25, 1:50, 1:75, 1:100, and 1:150 TA-CaCO_3_.

### 3.3. Characterization of TA-CaCO_3_

For SEM analysis, the prepared TA-CaCO_3_ materials were coated with platinum. Then, the morphologies and sizes of the as-prepared TA-CaCO_3_ materials were examined using field-emission scanning electron microscopy (FE-SEM, S-4700, Hitachi, Japan). For size analysis of the prepared TA-CaCO_3_, randomly selected individual particles of each TA-CaCO_3_ type were analyzed using ImageJ software (Version 1.47; US National Institutes of Health, Bethesda, MD, USA). The surface elemental composition of TA-CaCO_3_ was examined using SEM-EDS.

To determine the amount of Ca^2+^ in TA-CaCO_3_, 1:75 TA-CaCO_3_ materials (0.1 mg) were co-treated with nitric acid and H_2_O_2_, and the prepared samples were analyzed using ICP-OES (Optima 7300 DV, PerkinElmer, Waltham, MA, USA). Through ICP-OES analysis, the amount of TA and CaCO_3_ within TA-CaCO_3_ was determined using a standard curve (Y = 10,807x + 1340.9; R^2^ = 0.999).

The prepared TA-CaCO_3_ materials were characterized using FT-IR (Shimadzu 8400S, Kyoto, Japan). The FT-IR spectrum was acquired using a KBr pellet at a resolution of 4 cm^−1^ between 4,000 and 400 cm^−1^.

XRD and XPS analyses of 1:75 TA-CaCO_3_ were done by using a K-alpha+ (Thermo Scientific, Waltham, MA, USA) with Cu Kα radiation and a D8 ADVANCE diffractometer (Bruker, Germany) with Cu Kα radiation, respectively.

### 3.4. Antacid Effects of TA-CaCO_3_ Materials

To investigate the in vitro antacid effects, CaCO_3_ (10 mg) or TA-CaCO_3_ (1:75, 10 mg) was dispersed in 2 mL of two different solutions, PBS (pH = 7.4) and SGF (pH = 1.5). Each solution (1.4 mL) was mixed well with the as-prepared BTB solution (0.6 mL) and stirred. After 1 h, each solution was centrifuged at 13,500 rpm for 30 min. The color of the collected supernatant was observed by taking photos, and the absorbance was analyzed using a UV/vis spectrophotometer (NEO-S490, NEOGEN, Daejeon, Korea).

### 3.5. Antioxidant Effects of TA-CaCO_3_ Materials

To examine and compare the in vitro antioxidant effects, CaCO_3_ (1 mg) or TA-CaCO_3_ (1 mg) was added to 1 mL of DPPH (0.1 mM) dissolved in MeOH and reacted at 600 rpm for 30 min. Then, each solution was centrifuged at 5,500 rpm for 10 min. The color of the supernatant was observed by taking photos, and the absorbance was recorded using a UV/vis spectrophotometer (NEO-S490).

### 3.6. In Vitro Cytotoxicity and Anti-Inflammatory Effects

Human knee articular chondrocytes were obtained from Lonza (Basel, Switzerland) and cultured in low-glucose Dulbecco’s modified Eagle’s medium (DMEM, Welgene, Seoul, Korea), supplemented with 10% fetal bovine serum (FBS) and 1% penicillin-streptomycin at 37 °C.

The cytotoxicity of TA-CaCO_3_ materials was determined with a cell counting kit-8 (CCK-8, Dojindo, Kumamoto, Japan). Chondrocytes (1 × 10^4^ cells/well) were seeded and incubated in a 96-well plate. After 24 h of incubation, TA-CaCO_3_ materials at 0, 10, 50, and 100 μg/mL concentrations were treated into the cells. After 24 or 48 h of exposure, the CCK-8 reagent was additionally treated to the cells for 1 h, and the optical density was then recorded at 450 nm using a Multimode Reader. The cell viability of each group was represented as the percentage of viable cells versus the control group.

The in vitro anti-inflammatory effects of TA-CaCO_3_ against LPS-stimulated chondrocytes were evaluated by determining the mRNA levels of pro-inflammatory factors using a real-time polymerase chain reaction (PCR). Cells (1 × 10^5^ cells/well) were seeded into 24-well plates and cultured overnight. Cells were simultaneously treated with both LPS (100 ng/mL) and different concentrations of TA-CaCO_3_ (0, 10, 50, and 100 μg/mL). After three-day treatment, the cells from each group were collected, and the total RNA was extracted using the RNeasy Plus Mini Kit (Qiagen, Valencia, CA, USA) according to the manufacturer’s guidelines. Total RNA (1 μg) was reverse-transcribed into cDNA using AccuPower RT PreMix (Bioneer, Daejeon, Korea). The sequences of the target gene primers are provided in Appendix A (Appendix A). Real-time PCR analysis was conducted using an ABI7300 Real-Time Thermal Cycler (Applied Biosystems, Foster City, CA, USA). The mRNA levels of the target genes were normalized to those of glyceraldehyde 3-phosphate dehydrogenase (GAPDH) and presented as relative levels.

### 3.7. Antioxidant Effects of TA-CaCO_3_ at the Cellular Level

To investigate the antioxidant effects of TA-CaCO_3_ at the cell level, we conducted DCFDA staining and a DCFDA assay. Cells (1 × 10^4^ cells/well) were seeded and incubated in 24-well plates with microscope cover glasses for 24 h. Meanwhile, the extract solutions from the different concentrations of TA-CaCO_3_ (0, 10, 50, and 100 μg/mL) were prepared by incubating them in DMEM medium at 37 °C for 24 h, followed by collecting the supernatants from each group. Next, cells were treated with 300 μM of H_2_O_2_ at 37 °C for 30 min, followed by additional treatment of the collected extract solutions from each group for 24 h. After washing the cells with PBS, the cells were stained with DCFDA (25 μM) for 30 min under a dark condition, washed with PBS, and then fixed with 3.7% paraformaldehyde for 30 min. The fluorescence intensity within the cells was observed using a confocal laser scanning microscope (CLSM, LSM700, Zeiss, Germany).

### 3.8. Protection of Cell Viability in the ROS Environment

To investigate whether TA-CaCO_3_ can protect cell viability in an ROS environment, chondrocytes (1 × 10^5^ cells/well) were seeded in a 24-well culture plate. Then, the cells were treated with 300 μM of H_2_O_2_ at 37 °C for 30 min, followed by additional treatment of TA-CaCO_3_ at 10, 50, and 100 μg/mL concentrations. After 24 h of treatment, the CCK-8 (Dojindo) reagent was added to the cells for 1 h. The supernatant from each group was transferred into a 96-well plate, and its optical density was measured at 450 nm using a multimode reader.

## 4. Conclusions

In the present study, we prepared TA-CaCO_3_ materials by reacting TA with CaCl_2_ and Na_2_CO_3_, which led to the interaction between TA and Ca^2+^ ions, followed by nucleation of CaCO_3_. Micron-sized 1:75 TA-CaCO_3_ materials (ranging from 3 to 6 μm) comprised small nanoparticles in a size range of 17–41 nm. TA-CaCO_3_ materials could effectively neutralize the SGF solution and scavenge free radicals. In addition, these particles significantly suppressed the mRNA expression of pro-inflammatory cytokines and mediators and scavenged intracellular ROS in cells. Their anti-inflammatory and antioxidant activities protected chondrocytes from ROS. These results suggest that TA-CaCO_3_ materials have excellent antacid, antioxidant, and anti-inflammatory properties. Importantly, TA molecules can undergo multiple interactions with nucleic acids, peptides, proteins, and polysaccharides. Furthermore, due to the molecular adsorption of CaCO_3_ materials, CaCO_3_-based materials can improve the incorporation efficacy of drugs. Thus, using TA-CaCO_3_ materials, we will develop dual drug delivery systems that can ferry both a chemical drug and protein drug, and then apply them to treat inflammatory cells or diseases.

## Figures and Tables

**Figure 1 ijms-22-04614-f001:**
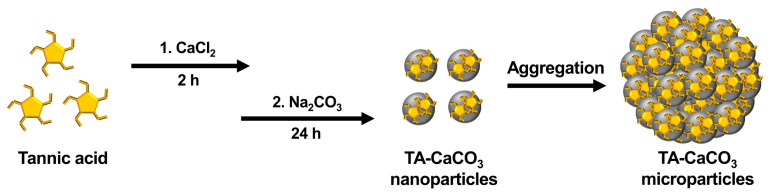
Schematic illustration of the synthesis of TA-CaCO_3_ materials. TA was sequentially reacted with CaCl_2_ for 2 h and Na_2_CO_3_ for an additional 24 h in pure water (pH = 7.0).

**Figure 2 ijms-22-04614-f002:**
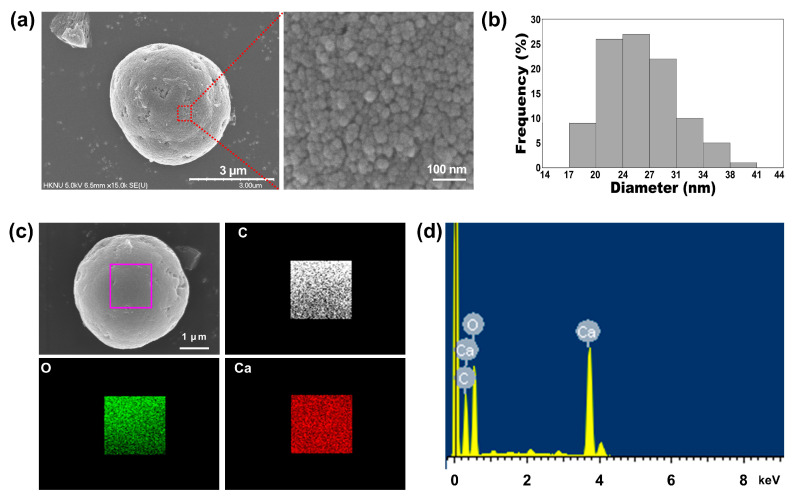
(**a**) The representative SEM images and magnified surface of 1:75 TA-CaCO_3_ materials. The magnified SEM image revealed that the micron-sized TA-CaCO_3_ materials comprised small TA-CaCO_3_ nanoparticles. (**b**) Particle size distribution of the small TA-CaCO_3_ nanoparticles of 1:75 TA-CaCO_3_. (**c**) EDS mapping of elemental carbon (C; white), oxygen (O; green), and calcium (Ca; red). Scale bar: 1 μm. (**d**) EDS spectrum of 1:75 TA-CaCO_3_.

**Figure 3 ijms-22-04614-f003:**
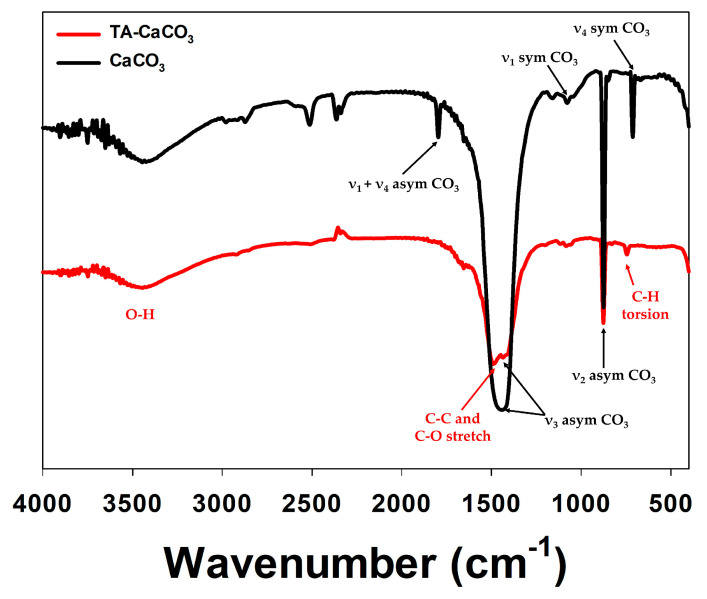
FT-IR spectra of commercial CaCO_3_ and TA-CaCO_3_.

**Figure 4 ijms-22-04614-f004:**
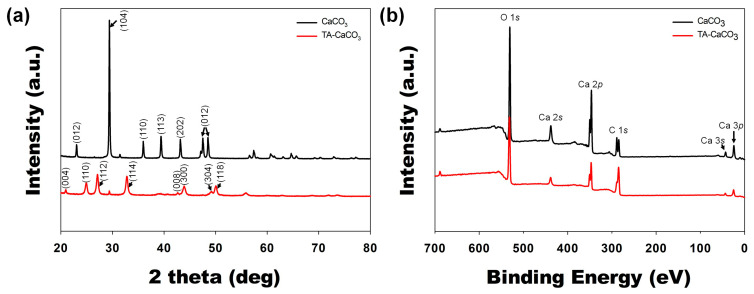
XRD and XPS analyses of commercial CaCO_3_ and TA-CaCO_3_ materials. (**a**) XRD spectra of CaCO_3_ and TA-CaCO_3_. (**b**) Wide scan XPS spectra recorded from CaCO_3_ and TA-CaCO_3_.

**Figure 5 ijms-22-04614-f005:**
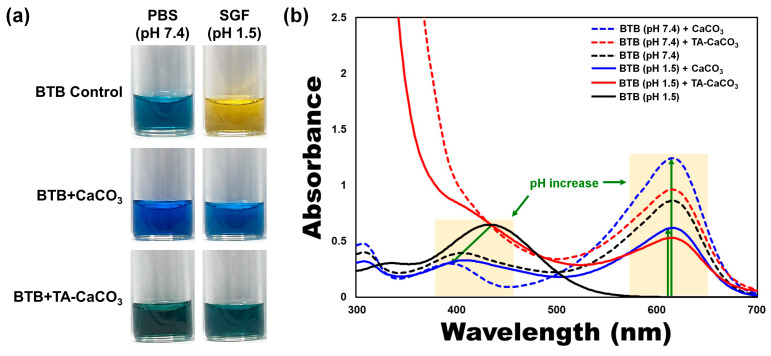
Antacid effects of TA-CaCO_3_ using the colorimetric bromothymol blue (BTB) method. (**a**) Color changes and (**b**) UV/vis spectra of BTB under PBS (pH = 7.4) and simulated gastric fluid (SGF, pH = 1.5) after treatment with commercial CaCO_3_ and 1:75 TA-CaCO_3_. Green-colored arrows indicate a pH increase.

**Figure 6 ijms-22-04614-f006:**
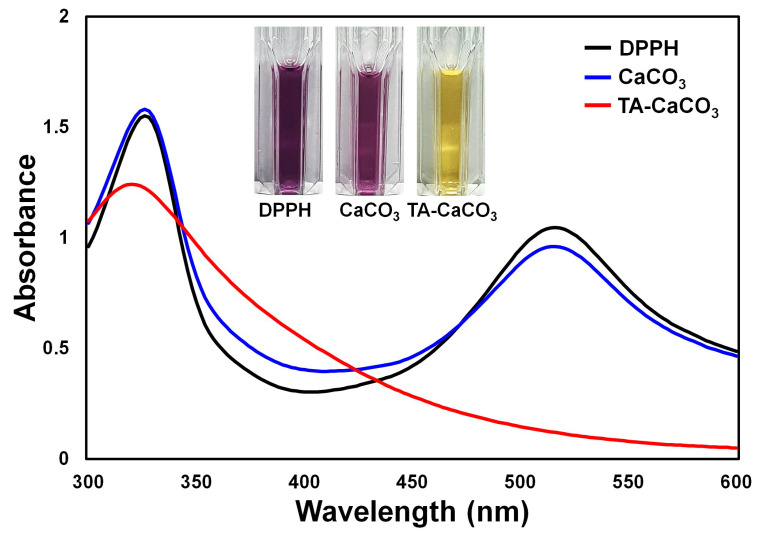
Antioxidant effects of TA-CaCO_3_ determined using the colorimetric DPPH method. Changes in the color and UV/vis spectra of DPPH after treatment with commercial CaCO_3_ and 1:75 TA-CaCO_3_. Inset: Photos of the DPPH solution treated with commercial CaCO_3_ and 1:75 TA-CaCO_3_.

**Figure 7 ijms-22-04614-f007:**
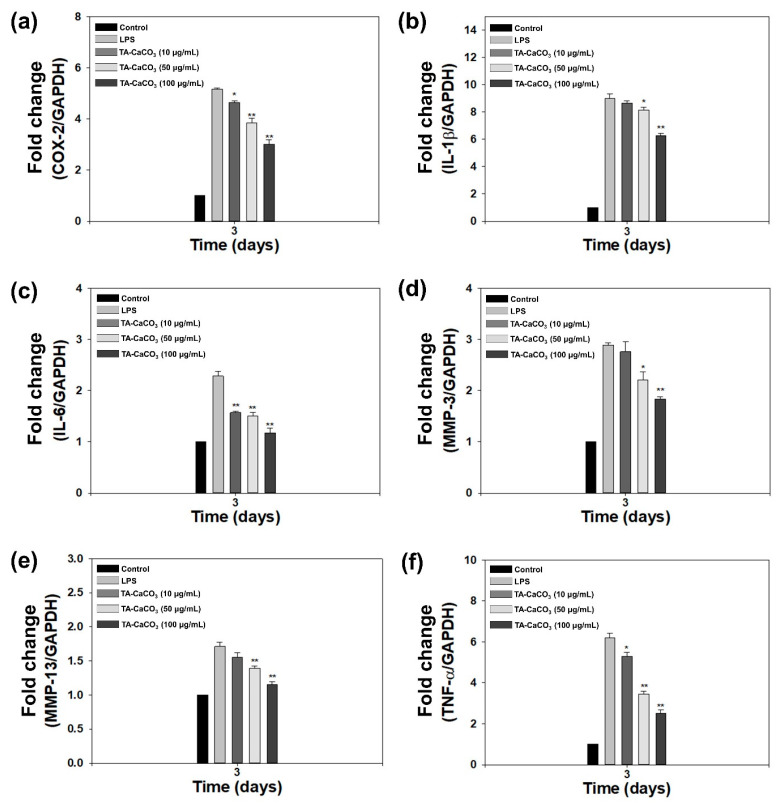
In vitro anti-inflammatory effects of 1:75 TA-CaCO_3_ in inflamed chondrocytes. The mRNA levels of pro-inflammatory factors, including (**a**) cyclooxygenase-2 (COX-2), (**b**) interleukin-1β (IL-1β), (**c**) IL-6, (**d**) matrix metalloproteinase-3 (MMP-3), (**e**) MMP-13, and (**f**) tumor necrosis factor-α (TNF-α) in LPS-stimulated chondrocytes on day three. Data are represented as the mean ± SD (*n* = 5). ** *p* < 0.01 and * *p* < 0.05.

**Figure 8 ijms-22-04614-f008:**
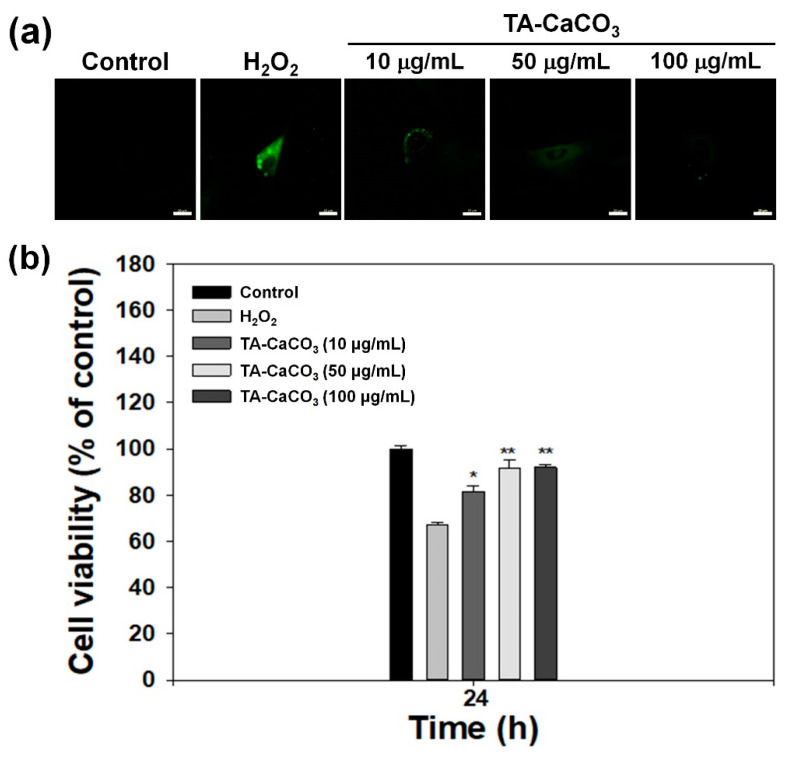
In vitro antioxidant effects of TA-CaCO_3_. (**a**) Representative fluorescence images of intracellular ROS in chondrocytes treated with extracts containing different concentrations of 1:75 TA-CaCO_3_ for 24 h after pretreatment with 300 μM of H_2_O_2_ for 30 min. Scale bar: 20 μm. (**b**) Viability of chondrocytes treated with different concentrations of 1:75 TA-CaCO_3_ for 24 h after pretreatment with 300 μM of H_2_O_2_ for 30 min. Results are expressed as the mean ± SD (*n* = 5). ** *p* < 0.01 and * *p* < 0.05.

## Data Availability

Not applicable.

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
