# Peer review of "Tannylated Calcium Carbonate Materials with Antacid, Anti-Inflammatory, and Antioxidant Effects"

_ijms, 2021, doi:10.3390/ijms22094614_

Round 1
Reviewer 1 Report
The authors synthesized the tannylated CaCO3 (TA-CaCO3) materials using a simple reaction between tannic acid (TA) and calcium and carbonate ions, and studied the abiotic antacid and antioxidant effects, as well as the anti-inflammatory and antioxidant properties in vitro using the chondrocytes. However, the reviewer find that the design of in vitro experiments is not clear, and some control experiments are missing. In addition, it will also be interesting to demonstrate the antacid effects of the materials in vitro using a cancer cell line, as tumor microenvironment is acidic. The antacid property might lead to an anti-tumor effect. The potential application of the materials needs to be discussed in the manuscript. Some detailed comments to consider:
Page 3 Figure 2d Why there are two spectra of calcium by EDS analysis?
Page 3 Line 104 The figure of ICP-OES results is missing.
Page 4 Figure 4 The labels of the curves are confusing. The current SGF and PBS curves are the absorbances of BTB under different conditions. The actual baselines of SGF and PBS alone are missing here. The authors can relabel the current samples as: BTB (pH 7.4), CaCO3+BTB (pH 1.5), TA-CaCO3+BTB (pH 1.5), and BTB (pH 1.5). The absorbances of CaCO3+BTB and TA-CaCO3+BTB under pH 7.4 can be plotted in the supplemental figure.
Page 5 Line 136-140 The presence of CaCO3 shifts the absorption band of BTB from 433 to 615 nm, suggesting the neutralization of the solution, but why is the band at 433 nm in CaCO3+BTB group still higher than BTB alone?
Page 6 Why the authors use the chondrocytes? Is it a murine cell line? The cytotoxicity of TA-CaCO3 materials towards the cells need to be investigated first. For the LPS stimulated experiments, it is not clear how the experiments were performed. How long were the cells pre-treated with LPS? Were the LPS removed before adding the materials? These need to be clarified in the Method section (Page 9, Line 272). In addition, will it be possible that the anti-inflammatory effects are simply due to the recovery of the cell viability?
Page 9 Again, the experiment design of the antioxidant effects is not clear.
Line 286 Why the pre-treated cells with cultured without FBS?
Line 291 It seems that the cells were exposed to the materials first, and then challenged with H2O2. What happened when the cells treated first with the materials? I assume there was no oxidative stress at that time. Why the cells protected the cell viability afterwards when exposed to H2O2?
Author Response
Responses to Reviewer-1’s comments
The authors synthesized the tannylated CaCO3 (TA-CaCO3) materials using a simple reaction between tannic acid (TA) and calcium and carbonate ions, and studied the abiotic antacid and antioxidant effects, as well as the anti-inflammatory and antioxidant properties in vitro using the chondrocytes. However, the reviewer find that the design of in vitro experiments is not clear, and some control experiments are missing. In addition, it will also be interesting to demonstrate the antacid effects of the materials in vitro using a cancer cell line, as tumor microenvironment is acidic. The antacid property might lead to an anti-tumor effect. The potential application of the materials needs to be discussed in the manuscript. Some detailed comments to consider:
Answer: We really appreciate the time and effort that you and the reviewers have dedicated to the timely review of our manuscript. Our group has made substantive changes to the revised manuscript in accordance with the comments or suggestions of the reviewers. These changes have made a positive impact on the clarity and technical content of the paper.
We address the Reviewers’ comments, our answers, and the changes of the revised version according to the comments or suggestions of the reviewers. The original comments are shown in black and our answers are given in blue. The appropriate changes made in the revised manuscript are red.
Comment-1: Page 3 Figure 2d Why there are two spectra of calcium by EDS analysis?
Answer: Below figure is an example of how EDS works. The letters K, L, and M refer to the n value that electrons in that shell have (K electrons, closest to the nucleus, are n=1 electrons), while α and β indicate the size of the transition. The relaxation from M to L or L to K are therefore described as Lα or Kα, while going from M to K would be a Kβ transition. The means that are used for describing these processes as a whole are known as Siegbahn notation.
Also, website (http://www.edax,com/resources/interactive-periodic-table) provides the peaks of transitions such as Kα, Lα, and Kβ transitions corresponding to each atom. According to website, calcium showed two energy peaks of Kα (3.691 eV) and Lα (0.341 eV). To clarify EDS analysis, we added the sentence for EDS analysis.
Addition (Line 108~110): Consistent with previous report [29], EDS spectrum showed the peaks of Kα (0.277 eV) corresponding to carbon, Kα (0.523 eV) corresponding to oxygen, and Kα (3.691 eV) and Lα (0.341 eV) corresponding to calcium.
Added reference: [29] Gokhe, U. B.; Koparkar, K. A.; Omanwar, S. K., Synthesis and fluorescence properties of Ca2SiO4:Dy3+ phosphor for solid state lighting application. J Mater Sci Mater Electron 2016, 27, 9286-9290.
Comment-2: Page 3 Line 104 The figure of ICP-OES results is missing.
Answer: Thank you for the comment. We just calculated the amounts of CaCO3 and TA within 0.1 mg of 1:75 TA-CaCO3 based on the standard curve (Y = 10,807x + 1340.9; R2=0.999) after performing ICP-OES analysis. Thus, we just described their amounts in the text. To clarify ICP-OES result, we revised the sentence.
Correction (Line 227~279): Through ICP-OES analysis, the amount of TA and CaCO3 within TA-CaCO3 was determined using a standard curve (Y = 10,807x + 1340.9; R2=0.999).
Comment-3 and Comment-4: Page 4 Figure 4 The labels of the curves are confusing. The current SGF and PBS curves are the absorbances of BTB under different conditions. The actual baselines of SGF and PBS alone are missing here. The authors can relabel the current samples as: BTB (pH 7.4), CaCO3+BTB (pH 1.5), TA-CaCO3+BTB (pH 1.5), and BTB (pH 1.5). The absorbances of CaCO3+BTB and TA-CaCO3+BTB under pH 7.4 can be plotted in the supplemental figure. Page 5 Line 136-140 The presence of CaCO3 shifts the absorption band of BTB from 433 to 615 nm, suggesting the neutralization of the solution, but why is the band at 433 nm in CaCO3+BTB group still higher than BTB alone?
Answer: Thank you for these two comments. As reviewer suggested, we relabeled the Figure 4. Also, we found that there are some errors for Figure 4. Thus, we performed in vitro antacid experiments by slightly modifying the protocols and then revised Figure 4 data and the description in the text.
Correction (Line 155~165): CaCO3 and TA-CaCO3 showed a deep blue color of BTB at pH 7.4, indicating the slight pH increases of the solution following the degradation of CaCO3 and TA-CaCO3 even at pH 7.4. In contrast, the yellowish BTB solution at pH 1.5 turned to bluish green color after reaction with CaCO3 and TA-CaCO3, indicating that the pH of the solution increased to a nearly neutral pH (approximately 7). Consistent with these color changes, the λmax shift of BTB occurred from 615 nm at pH 7.4 to 433 nm under SGF (pH 1.5) (Figure 5b). However, the absorptions of both CaCO3 and TA-CaCO3 groups increased at 615 nm, while the λmax of BTB at pH 1.5 was blue-shifted from 433 nm to 403 nm, implying the pH increases of the solutions after reacting with two kinds of CaCO3 materials. Meanwhile, TA-CaCO3 showed significantly higher absorbance below 400 nm at both pH 1.5 and pH 7.4, meaning the presence of TA in the solutions.
Corrected Figure 4:
Figure 4. Antacid effects of TA-CaCO3 using the colorimetric bromothymol blue (BTB) method. (a) Color changes and (b) UV/Vis spectra of BTB under PBS (pH 7.4) and simulated gastric fluid (SGF, pH 1.5) after treatment with commercial CaCO3 and 1:75 TA-CaCO3. Green-colored arrows indicate pH increase.
Correction (Line 287~290): To investigate the in vitro antacid effects, CaCO3 (10 mg) or TA-CaCO₃ (1:75, 10 mg) were dispersed in 2 mL of two different solutions, PBS (pH 7.4) and SGF (pH 1.5). Each solution (1.4 mL) was mixed well with the as-prepared BTB solution (0.6 mL) and stirred. After 1 h, each solution was centrifuged at 13,500 rpm for 30 min.
Comment-5: Page 6 Why the authors use the chondrocytes? Is it a murine cell line? The cytotoxicity of TA-CaCO3 materials towards the cells need to be investigated first. For the LPS stimulated experiments, it is not clear how the experiments were performed. How long were the cells pre-treated with LPS? Were the LPS removed before adding the materials? These need to be clarified in the Method section (Page 9, Line 272). In addition, will it be possible that the anti-inflammatory effects are simply due to the recovery of the cell viability?
Answer: Thank you for the comment. In this study, we used human knee articular chondrocytes which is purchased from Lonza. As reviewer suggested, we performed in vitro cytotoxicity study using CCK-8 kit and provided cytotoxicity result as Figure S3 in the Supplementary Materials. Additionally, as reviewer indicated, we found that the experiment protocol for in vitro anti-inflammatory study is not clear. Thus, we revised the sentences to clarify in vitro anti-inflammatory study. Moreover, the recovery of the cell viability might be associated with the anti-inflammatory effects. As reviewer also suggested in comment-8, we discussed this issue in the text.
Correction (Line 299~300):
3.6. In Vitro Cytotoxicity and Anti-inflammatory Effects
Human knee articular chondrocytes were obtained from Lonza (Basel, Switzerland) ~
Addition (Line 186~188): Cytotoxicity study of TA-CaCO3 was examined against normal chondrocytes. Figure S3 revealed that chondrocytes maintained their cell viabilities above 90% up to 100 μg/mL concentration, indicating that TA-CaCO3 is non-toxic.
Figure S3. In vitro cytotoxicity of 1:75 TA-CaCO3 against chondrocytes.
Addition (Line 304~310): Cytotoxicity of TA-CaCO3 materials was determined with cell counting kit-8 (CCK-8, Dojindo, Kumamoto, Japan). Chondrocytes (1 ´ 104 cells/well) were seeded and incubated in a 96-well plate. After 24 h incubation, TA-CaCO3 materials at 0, 10, 50, and 100 μg/mL concentrations were treated into the cells. After 24 or 48 h exposure, CCK-8 reagent was additionally treated to the cells for 1 h, and the optical density was then recorded at 450 nm using a Multimode Reader. The cell viability of each group was represented as the percentage of viable cells versus the control group.
Correction (Line 314~316): Cells were simultaneously treated with both LPS (100 ng/mL) and different concentrations of TA-CaCO3 (0, 10, 50, and 100 mg/mL). After 3-day treatment, the cells from each group ~
Addition (Line 361~362): Figure S3: In vitro cytotoxicity of 1:75 TA-CaCO3 against chondrocytes.
Comment-6: Page 9 Again, the experiment design of the antioxidant effects is not clear.
Answer: Thank you for the comment. As reviewer indicated, we found that the experiment protocol for in vitro antioxidant is not clear. To clarify, we revised the experimental method.
Correction (Line 327~332): Meanwhile, the extract solutions from the different concentrations of TA-CaCO3 (0, 10, 50, and 100 mg/mL) were prepared by incubating them in DMEM medium at 37°C for 24 h, followed by collecting the supernatants from each group. Next, cells were treated with 300 mM H2O2 at 37°C for 30 min, followed by additional treatment of the collected extract solutions from each group for 24 h. After washing the cells with PBS, the cells were stained with DCFDA ~
Comment-7: Line 286 Why the pre-treated cells with cultured without FBS?
Answer: We made a mistake, and it is a typo. Also, we revised the experimental protocols regarding antioxidant study. See the revised experiments in the above response to comment-6.
Comment-8: Line 291 It seems that the cells were exposed to the materials first, and then challenged with H2O2. What happened when the cells treated first with the materials? I assume there was no oxidative stress at that time. Why the cells protected the cell viability afterwards when exposed to H2O2?
Answer: Thank you for the comment. As reviewer indicated, we also found that the experiment protocol for the protection of cell viability is something wrong. To induce oxidative stress, the cells were treated with H2O2 for 30 min. Then, TA-CaCO3 materials at different concentrations were additionally treated. Thus, we corrected the experimental method. Moreover, the protection of cell viability for TA-CaCO3 materials was already discussed in the text.
Correction (Line 338~341): Then, the cells were treated with 300 mM H2O2 at 37°C for 30 min, followed by additional treatment of TA-CaCO3 at 10, 50, and 100 mg/mL concentrations. After 24 h treatment, CCK-8 (Dojindo) reagent was added to the cells for 1 h.

Reviewer 2 Report
The reviewed work presents the results of research conducted on the creation of a biomaterial in the form of microparticles, which are conglomerates of calcium carbonate, calcium chloride and tannic acid, showing anti-acid, antioxidant and anti-inflammatory properties.
The work is essentially interesting, it seems that it can be published, but it is imperative that the authors should clarify some important issues before.
- Can the authors attach the results confirming the assumed mechanism of conglomerate formation? In the reaction of tannic acid with divalent cations ions, formation of a water-insoluble complex may take place. It is required to maintain an appropriate regime of reaction conditions - mainly to maintain an appropriate pH. This is missing in the description of the conducted reaction.
- The work does not sufficiently define the composition and structure of the microparticles formed. This is very important from the point of view of the planned biomedical application of the produced material. The data obtained from the study of the morphology and surface of microparticles are not enough. FTIR measurement results are also not convincing. It is not really in detail known how they are made and what these synthesized particles are made of. Measurement results are needed for this characteristic; XRD, DSC. When expanding the description of conglomerate formation and the results of its research, please take into account the previous observations concerning the formation of similar particles in the reaction of calcium chloride and sodium carbonate, described in detail in the article; Composition inversion to form calcium carbonate mixtures, CrystEngComm, 2017, 19, 3573 -3583.
- The lack of more detailed characteristics of the obtained material is also related to the lack of clearly formulated conclusions regarding the influence of synthesis conditions on the final properties of the final conglomerate. Without these data, the possibility of obtaining adequate repeatability of the tested process is quite doubtful. Has the reproducibility of the microparticle synthesis method been confirmed? Authors should also supplement the manuscript with;
- In introduction part, it is necessary to mention the requirements related to the chemical purity of carbonates used in medicine. Describe please the problems with obtaining synthetic carbonate of appropriate purity to maintain biocompatibility, and cite several works on this topic.
- in the Conclusion section, the authors should present a vision of the intended use and application of the obtained material in medicine.
- it is necessary to improve the quality of the some graphics and drawings
Author Response
Responses to Reviewer-2’s comments
The reviewed work presents the results of research conducted on the creation of a biomaterial in the form of microparticles, which are conglomerates of calcium carbonate, calcium chloride and tannic acid, showing anti-acid, antioxidant and anti-inflammatory properties.
The work is essentially interesting, it seems that it can be published, but it is imperative that the authors should clarify some important issues before.
Answer: We really appreciate the time and effort that you and the reviewers have dedicated to the timely review of our manuscript. Our group has made substantive changes to the revised manuscript in accordance with the comments or suggestions of the reviewers. These changes have made a positive impact on the clarity and technical content of the paper.
We address the Reviewers’ comments, our answers, and the changes of the revised version according to the comments or suggestions of the reviewers. The original comments are shown in black and our answers are given in blue. The appropriate changes made in the revised manuscript are red.
Comment-1: Can the authors attach the results confirming the assumed mechanism of conglomerate formation? In the reaction of tannic acid with divalent cations ions, formation of a water-insoluble complex may take place. It is required to maintain an appropriate regime of reaction conditions - mainly to maintain an appropriate pH. This is missing in the description of the conducted reaction.
Answer: Thank you for the comment. As reviewer suggested, we revised Figure 1 to confirm the assumed mechanism of conglomerate formation. Also, we added the reaction condition which is conducted in pure-water (pH 7.0) and revised the legend of Figure 1.
Correction (Line 70~72): TA interacts with Ca2+ ions and facilitates the nucleation of CaCO3 to form TA-CaCO3 nanoparticles, which then agglomerates into microparticles due to the interactions between TA-CaCO3 nanoparticles.
Revised Figure 1 and legend:
Figure 1. Schematic illustration of the synthesis of TA-CaCO3 materials. TA was sequentially reacted with CaCl2 for 2 h and Na2CO3 for additional 24 h in pure water (pH 7.0).
Comment-2: The work does not sufficiently define the composition and structure of the microparticles formed. This is very important from the point of view of the planned biomedical application of the produced material. The data obtained from the study of the morphology and surface of microparticles are not enough. FTIR measurement results are also not convincing. It is not really in detail known how they are made and what these synthesized particles are made of. Measurement results are needed for this characteristic; XRD, DSC. When expanding the description of conglomerate formation and the results of its research, please take into account the previous observations concerning the formation of similar particles in the reaction of calcium chloride and sodium carbonate, described in detail in the article; Composition inversion to form calcium carbonate mixtures, CrystEngComm, 2017, 19, 3573 -3583.
Answer: Thank you for the comment. As reviewer suggested, FT-IR data is not enough to convince the synthesis of TA-CaCO3 materials. As reviewer recommended, therefore, we additionally characterized the prepared TA-CaCO3 materials with XRD and XPS analyses, and we further described and discussed the obtained XRD and XPS data in the manuscript.
Also, we really appreciate for suggesting the reference. However, the synthesis condition of CaCO3 materials in recommenced reference is quite different from our synthetic method for TA-CaCO3. Due to the different synthesis method, we think the comparison of the prepared materials is not appropriate.
Correction (Line 103~106): The preparation of TA-CaCO3 materials was confirmed using a SEM coupled with energy-dispersive (SEM-EDS) X-ray spectroscopy, inductively coupled plasma optical emission spectrometry (ICP-OES), Fourier-transform infrared (FT-IR) spectroscopy, X-ray diffraction (XRD) patterns, and X-ray photoelectron spectroscopy (XPS).
Addition (Line 122~129): Next, commercial CaCO3 and synthesized TA-CaCO3 crystal phases were identified via XRD analysis (Figure 4a). CaCO3 exhibited the characteristic peaks such as the plane of the calcite at 29.3° (104), and the calcite crystal faces at 23.02° (012), 35.9° (110), 39.4° (113), and 43.1°(202), respectively. Meanwhile, the diffraction peaks of TA-CaCO3 were observed at 24.8° (110), 27.08° (112), 32.7° (114), 43.8° (300), 49.1° (304), and 50.08° (118), respectively, and these diffraction peaks are consistent with the vaterite crystal faces [33, 34]. Based on XRD data, we think that the prepared TA-CaCO3 materials are the vaterite form of calcium carbonate.
Added references:
- Wojtas, M.; Wołcyrz, M.; Ożyhar, A.; Dobryszycki, P., Phosphorylation of Intrinsically Disordered Starmaker Protein Increases Its Ability To Control the Formation of Calcium Carbonate Crystals. Cryst Growth Des 2012, 12, (1), 158-168.
- Dong, W.; Tu, C.; Tao, W.; Zhou, Y.; Tong, G.; Zheng, Y.; Li, Y.; Yan, D., Influence of the Mole Ratio of the Interacting to the Stabilizing Portion (RI/S) in Hyperbranched Polymers on CaCO3 Crystallization: Synthesis of Highly Monodisperse Microspheres. Cryst Growth Des 2012, 12, (8), 4053-4059.
Addition (Line 133~1137): XPS data revealed that CaCO3 and TA-CaCO3 showed Ca, O, and C signals (Figure 4b). The binding energy peaks of these two materials appeared O1s at 531 eV, Ca2s at 441 eV, two Ca2p at 351 and 347.2 eV, and two C1s peaks at 289.3 eV (CO3 in the CaCO3 surface) and 284.6 eV (adventitious carbon peak), respectively. These data demonstrated the successful synthesis of TA-CaCO3 and the synthesized TA-CaCO3 materials are vaterite calcium carbonate.
Added new data (Figure 4) regarding XRD and XPS:
Figure 4. XRD and XPS analyses of commercial CaCO3 and TA-CaCO3 materials. (a) XRD spectra of CaCO3 and TA-CaCO3. (b) Wide scan XPS spectra recorded from CaCO3 and TA-CaCO3.
Comment-3: The lack of more detailed characteristics of the obtained material is also related to the lack of clearly formulated conclusions regarding the influence of synthesis conditions on the final properties of the final conglomerate. Without these data, the possibility of obtaining adequate repeatability of the tested process is quite doubtful. Has the reproducibility of the microparticle synthesis method been confirmed? Authors should also supplement the manuscript with;
Answer: Thank you for the comment. As shown in the response to above comment-2, we performed additional characteristics of the materials. In our present study, particularly, we provided additional data to supplement the lack of the characteristics of 1:75 TA-CaCO3 by showing XPS and XRD data. Also, reviewer indicated the reproducibility of the microparticle synthesis method. Honestly, we performed the synthesis of TA-CaCO3 microparticles under the same synthetic conditions such as same amounts of the chemicals, same reactors, same volume, same stirring rpm of reaction solution using same magnetic bars, etc. Also, through the repeated synthesis of the TA-CaCO3 materials under same synthetic conditions, we set up the synthetic protocols and performed our study.
Comment-4: In introduction part, it is necessary to mention the requirements related to the chemical purity of carbonates used in medicine. Describe please the problems with obtaining synthetic carbonate of appropriate purity to maintain biocompatibility, and cite several works on this topic.
Answer: Thank you for the comment. Calcium carbonate has been used an antacid and it can be orally administered as various formulations such as tablet, chewable tablet, capsule, and liquid. However, they have different amount of CaCO3 with different purity. Thus, we cannot describe the chemical purity of calcium carbonates used in medicine. Due to this reason, we just describe the use of calcium carbonate and its formulations in the introduction part.
Correction (Line 31~32): Calcium carbonate (CaCO3), an inorganic biomineral, has been used as antacid agent. It can be orally administered as tablet, chewable tablet, capsule, and liquid.
Comment-5: in the Conclusion section, the authors should present a vision of the intended use and application of the obtained material in medicine.
Answer: Thank you for the comment. This comment is very important for our group. As reviewer recommended, we describe our vision of further biomedical applications for TA-CaCO3.
Addition (Line 353~358): Importantly, TA molecules can undergo multiple interactions with nucleic acids, peptides, proteins and polysaccharides. Also, due to the molecular adsorption of CaCO3 materials, CaCO3-based materials can improve the incorporation efficacy of drugs. Thus, using TA-CaCO3 materials, we will develop dual drug delivery systems which can deliver both chemical drug and protein drug and apply them to treat inflammatory arthritis.
Comment-6: it is necessary to improve the quality of the some graphics and drawings
Answer: Thank you for the comment. We tried to improve the quality of all graphics and drawings and then changed.

Round 2
Reviewer 1 Report
Comments from the reviewer have been addressed.
Reviewer 2 Report
The authors have practically responded to all my previous objections and made appropriate changes to the manuscript. The form of their responses was very clear, facilitating a reliable assessment of the reviewed work. I believe the manuscript is suitable for publication.